# Geometry-aware Generation of Adversarial and Cooperative Point Clouds

## Abstract

Recent studies show that machine learning models are vulnerable to adversarial examples. In 2D image domain, these examples are obtained by adding imperceptible noises to natural images. This paper studies adversarial generation of point clouds by learning to deform those approximating object surfaces of certain categories. As 2D manifolds embedded in the 3D Euclidean space, object surfaces enjoy the general properties of smoothness and fairness. We thus argue that in order to achieve imperceptible surface shape deformations, adversarial point clouds should have the same properties with similar degrees of smoothness/fairness to the benign ones, while being close to the benign ones as well when measured under certain distance metrics of point clouds. To this end, we propose a novel loss function to account for imperceptible, geometry-aware deformations of point clouds, and use the proposed loss in an adversarial objective to attack representative models of point set classifiers. Experiments show that our proposed method achieves stronger attacks than existing methods, without introduction of noticeable outliers and surface irregularities. In this work, we also investigate an opposite direction that learns to deform point clouds of object surfaces in the same geometry-aware, but cooperative manner. Cooperatively generated point clouds are more favored by machine learning models in terms of improved classification confidence or accuracy. We present experiments verifying that our proposed objective succeeds in learning cooperative shape deformations.

## 1 Introduction

The existence of adversarial examples shows the vulnerability of machine learning models, and triggers a great amount of research attention paid to either attacking and defense studies for safety-critical issues, or robustness analysis on machine learning models themselves. In existing literature, adversarial examples generally satisfy two key properties: visually imperceptible by humans and capable to mislead machine learning models. For 2D adversarial images (Szegedy et al., 2013; Carlini & Wagner, 2017; Madry et al., 2017), their imperceptible nature indicates less sensitivity by humans to the texture changes of the original images, which can be viewed as noises with small magnitudes anchored on RGB values of pixels.

Different from texture changes in adversarial images, adversarial point clouds as approximate shape representations of object surface demand imperceptible shape deformations. Heuristic manners of existing several works are taken to allow shape deformations in arbitrary directions for adversarial attack effects. Consequently, they produce adversarial examples that contain obvious point outliers (Xiang et al., 2019; Liu et al., 2019a), which can thus be easily defended by simple outlier removal method such as SOR (Zhou et al., 2018).

In this work, we aim to adversarially perturbing benign point clouds with awareness of the underlying geometric properties, which can be generally described as some sort of approximate geometric smoothness, *e.g.,* point-wise normals (Wang et al., 2018), curvatures (Bae, 2006), or geodesics for pairs of points (He et al., 2019), in view of the nature of object surface as 2-manifold embedded in the 3D space. To this end, we propose a novel geometry-aware loss in combination with the attack one. Our proposed geometry-aware loss is composed of three terms: the Chamfer Distance (CD) and Hausdorff Distance (HD) terms prevent geometric/topological changes of global shape via constraining point-wise perturbations, while the term encouraging consistency of local curvatures

between the adversarial and benign point clouds achieves smooth local surface deformations. Extensive experiments on the ModelNet40 dataset (Wu et al., 2015) can verify the imperceptibility and effectiveness of geometry-aware adversaries.

Our success in generating less noticeable point clouds inspires us to think of an opposite direction: is it possible to perturb raw point clouds in a cooperative, geometry-aware manner such that the resulting imperceptible perturbations of point clouds can improve classification performance with existing machine learning models? To the best of our knowledge, very few existing works are pursuing this direction, at least in the domain of 3D shape analysis. Technically, the objective of cooperative generation on point clouds is achieved by learning point perturbations in favor of either improved classification confidence or corrected class predictions. In view of their geometry-aware nature, our cooperative point clouds could be practically meaningful in terms of guiding design of certain objects such that they can be better perceived by in-built machine learning systems in physical world. Experiments on the ModelNet40 again verify our motivation on cooperative generation of point clouds.

## 2 RELATED WORKS

A number of adversarial attack algorithms have been proposed on 3D semantic analysis (Xiao et al., 2019; Liu et al., 2019a; Xiang et al., 2019; Yang et al., 2019; Zheng et al., 2018; Wicker & Kwiatkowska, 2019; Liu et al., 2019b). Beyond (Xiao et al., 2019) manipulating both texture and shape on the mesh level to attack image classifiers and detectors, most of adversarial algorithms on point clouds attack 3D classifiers via points attachment (Xiang et al., 2019; Yang et al., 2019), point detachment (Yang et al., 2019; Zheng et al., 2018; Wicker & Kwiatkowska, 2019) or point-wise perturbation (Xiang et al., 2019; Liu et al., 2019a; Yang et al., 2019; Liu et al., 2019b). Existing methods fail to exploit geometric properties during adversarial example generation, and thus have evident outliers and can be easily defended by SOR defense (Zhou et al., 2018), which encourage our adversaries (see Figure 2 and Table 1).

A number of deep models have been proposed for point-based surface reconstruction such as Point Set Generation (PSG) (Fan et al., 2017), the AtlasNet (Groueix et al., 2018b), and Deep Cascade Generation (DCG) (Wang et al., 2019a) from 2D images. These methods exploiting local mesh structure into point clouds generation have inspired our geometry-aware concept, but they are directly regressing from the latent vector encoded from images rather than point-wise deformations in the proposed method.

Data argumentation on point clouds such as jittering, rotation and random scale firstly introduced in (Qi et al., 2017a) shares similar concept as our cooperative deformations, both of which generate new data to boost classification. However, the key differences between our cooperative learning and data argumentation lie in two folds: 1) optimization based vs. simple physical processing; and 2) deformations on the underlying shape vs. point density changes. Simply put, the proposed cooperative deformation can be a promising learning paradigm, while data argumentation is an effective pre-processing step to mitigate sparse sample distributions.

## 3 METHODOLOGY

Our problem setting assumes the availability of a collection of point clouds in the input space $\mathcal{X}$, which are approximate shape representations of object surfaces of certain categories. Each point cloud $\mathcal{P} \in \mathcal{X}$ contains an orderless set of $n$ points $\{\boldsymbol{p}_i\}_{i=1}^n$, with the corresponding label $y \in \mathcal{Y}$ of object category, where any $\boldsymbol{p}$ denotes the coordinates $(x, y, z)$ in the 3D Euclidean space. In this work, we focus on machine learning models of 3D point set classification (Qi et al., 2017a;b; Wang et al., 2019b), which learn a classifier $f : \mathcal{X} \to \mathcal{Y}$ and expect $f(\mathcal{P}) = y$ for any input $\mathcal{P}$ with the true label $y$.

$\mathcal{P}$ is a discrete approximation of an object surface that satisfies the general properties of *smoothness* and *fairness* (Botsch et al., 2010), which concern with the continuity and variation of (partial) derivatives of a parametric surface function; or in a more intuitive way, the properties concern with the curvatures of local surface patches. Our objective in this work is to obtain from $\mathcal{P}$ a deformed point cloud $\mathcal{P}'$ via learning to perturb individual points $\{\boldsymbol{p}_i\}_{i=1}^n$ of $\mathcal{P}$. Since the deformation is

expected to be imperceptible by humans, we argue that $\mathcal{P}'$ should have the aforementioned surface properties with similar degrees of smoothness/fairness to local surface patches of $\mathcal{P}$, while being globally close to $\mathcal{P}$ as well, measured by certain distance metrics of point sets; otherwise, humans would notice either the global, possibly topological changes of part configuration of the object surface, or those of local surface details. We present shortly in section 3.1 our technical solutions to the above objective of *geometry-aware point perturbations*, and discuss how such solutions can be used to generate either *adversarial* or *cooperative* point clouds in sections 3.2 and 3.3 respectively.

## 3.1 GEOMETRY-AWARE POINT PERTURBATIONS

In 2D image domain, pixel-wise $l_p$-norms are usually used to constrain the noise magnitudes of adversarial examples (Carlini & Wagner, 2017; Madry et al., 2017). Due to the sharply different data nature of point clouds, we consider the following learning criteria to perturb points $\{\boldsymbol{p}_i\}_{i=1}^n$ of $\mathcal{P}$ in a geometry-aware manner. These criteria are to constrain the deformation magnitudes of the resulting $\mathcal{P}'$, while taking into account prevention of point outliers and smooth regularization of local point neighborhoods. Combined use of these criteria leads to point cloud deformations that are less noticeable by humans, as verified in the comparative experiments in section 4.

**Chamfer Distance** Given two point sets $\mathcal{P}$ and $\mathcal{P}'$, the Chamfer distance computes

$$C_{\texttt{Chamfer}}(\mathcal{P}', \mathcal{P}) = \frac{1}{n} \sum_{\boldsymbol{p}' \in \mathcal{P}'} \min_{\boldsymbol{p} \in \mathcal{P}} \|\boldsymbol{p}' - \boldsymbol{p}\|_2^2 + \frac{1}{n} \sum_{\boldsymbol{p} \in \mathcal{P}} \min_{\boldsymbol{p}' \in \mathcal{P}'} \|\boldsymbol{p} - \boldsymbol{p}'\|_2^2, \tag{1}$$

which shows that Chamfer distance is symmetric w.r.t. $\mathcal{P}$ and $\mathcal{P}'$. Although the Chamfer distance (1) is not a strict distance function, since the triangle inequality does not hold, it is popularly used in the recent literature of learning based 3D shape generation (Yang et al., 2018; Groueix et al., 2018a; Fan et al., 2017). It measures the distance between the two point sets by averaging over the individual deviation of any $\boldsymbol{p} \in \mathcal{P}$ from $\mathcal{P}'$ and that of any $\boldsymbol{p}' \in \mathcal{P}'$ from $\mathcal{P}$. However, Chamfer distance is less effective in prevention of outlier points in $\mathcal{P}'$, since a small portion of outliers in $\mathcal{P}'$ increases the distance (1) negligibly — one can intuitively think of outliers of a point cloud as those away from the object surface, represented by the point cloud, with relatively large distances. This shortcoming of Chamfer distance motivates us to additionally use the Hausdorff distance as introduced below.

**Hausdorff Distance** In this work, we use a non-symmetric Hausdorff distance between the two point sets $\mathcal{P}$ and $\mathcal{P}'$ that computes

$$C_{\texttt{Hausdorff}}(\mathcal{P}', \mathcal{P}) = \max_{\boldsymbol{p}' \in \mathcal{P}'} \min_{\boldsymbol{p} \in \mathcal{P}} \|\boldsymbol{p}' - \boldsymbol{p}\|_2^2, \tag{2}$$

since only the deformation of $\mathcal{P}'$ is concerned. As (2) indicates, the Hausdorff distance finds the largest one among the smallest distances of individual $\boldsymbol{p}' \in \mathcal{P}'$ from $\mathcal{P}$. It is thus sensitive to generation of outliers in the resulting $\mathcal{P}'$.

Distances computed by the functions (1) and (2) rely on those between the individual points $\{\boldsymbol{p}'_i\}_{i=1}^n$ and $\{\boldsymbol{p}_i\}_{i=1}^n$, which does not involve geometries of local surface patches associated with these individual points; consequently, the resulting $\mathcal{P}'$ could be close to $\mathcal{P}$ when measured by (1) and/or (2), but changes of geometric details at certain local patches could be clearly visible, causing failure to achieve the objective of imperceptible deformation.

**Consistency of Local Curvatures** Our way of achieving geometry-aware imperceptibility is to constrain the point cloud deformation such that local patches of the surface approximated by the resulting $\mathcal{P}'$ have curvatures whose magnitudes are similar to those of the corresponding patches of the surface approximated by $\mathcal{P}$. Since computations in this work are conducted on the discrete surface approximation of point clouds, we propose discrete notions that approximately characterize curvatures of local surface patches.

More specifically, for any point $\boldsymbol{p}' \in \mathcal{P}'$, we find its closest point $\boldsymbol{p} \in \mathcal{P}$ by $\boldsymbol{p} = \arg\min_{\boldsymbol{p} \in \mathcal{P}} \|\boldsymbol{p}' - \boldsymbol{p}\|_2$. There exist local point neighborhoods $\mathcal{N}'_{\boldsymbol{p}'} \subset \mathcal{P}'$ and $\mathcal{N}_{\boldsymbol{p}} \subset \mathcal{P}$ respectively associated with $\boldsymbol{p}'$ and $\boldsymbol{p}$, which are obtained in this work by searching $k$ nearest neighbors, suggesting $|\mathcal{N}'_{\boldsymbol{p}'}| = |\mathcal{N}_{\boldsymbol{p}}| = k$. To capture the local geometry of $\mathcal{N}_{\boldsymbol{p}}$, we rely on the following discrete notion

$$\kappa(\boldsymbol{p}; \mathcal{P}) = \frac{1}{k} \sum_{\boldsymbol{q} \in \mathcal{N}_{\boldsymbol{p}}} |\langle (\boldsymbol{q} - \boldsymbol{p})/\|\boldsymbol{q} - \boldsymbol{p}\|_2, \boldsymbol{n}_{\boldsymbol{p}} \rangle|, \tag{3}$$

where $\boldsymbol{n_p}$ denotes the unit normal vector of the surface at $\boldsymbol{p}$. The term (3) intuitively measures the averaged angles between the normal vector and the vector defined by pointing $\boldsymbol{p}$ towards each $\boldsymbol{q}$ of its neighboring points. Indeed, since the normal vector $\boldsymbol{n_p}$ is orthogonal to the tangent plane of the surface at $\boldsymbol{p}$, each inner product in (3) characterizes how the normals vary directionally in the local neighborhood $\mathcal{N_p}$, thus approximately measuring the local, directional curvature, and an average of $|\mathcal{N_p}|$ inner products in (3) approximately measures the local, mean curvature. Note that unit normal vector $\boldsymbol{n_p}$ in (3) can be computed from $\mathcal{N_p}$ via eigen-decomposition of the set $\mathcal{N}(\boldsymbol{p})$ (Hoppe et al., 1992). We compute $\kappa'(\boldsymbol{p}';\mathcal{P}')$ in the same way as (3), with a subtle difference that instead of computing $\boldsymbol{n}'_{\boldsymbol{p}'}$ from $\mathcal{N}'_{\boldsymbol{p}'}$, we directly use $\boldsymbol{n_p}$, i.e., the unit normal vector of the point in $\mathcal{P}$ that is closest to $\boldsymbol{p}'$, as a surrogate of $\boldsymbol{n}'_{\boldsymbol{p}'}$, since normal vectors of $\mathcal{P}$ can be pre-computed and efficiently retrieved during the deformation learning.

Given $\kappa'(\boldsymbol{p}';\mathcal{P}',\mathcal{P})$ and $\kappa(\boldsymbol{p};\mathcal{P})$, we use the following criterion to encourage the consistency of local geometries between any $\boldsymbol{p}' \in \mathcal{P}'$ and its closest point $\boldsymbol{p} \in \mathcal{P}$

$$C_{\texttt{Curvature}}(\mathcal{P}',\mathcal{P}) = \frac{1}{n} \sum_{\boldsymbol{p}' \in \mathcal{P}'} \|\kappa'(\boldsymbol{p}';\mathcal{P}',\mathcal{P}) - \kappa(\boldsymbol{p};\mathcal{P})\|_2^2 \ \ \text{s.t.} \ \ \boldsymbol{p} = \arg\min_{\boldsymbol{p} \in \mathcal{P}} \|\boldsymbol{p}' - \boldsymbol{p}\|_2.$$
(4)

where we write $\kappa'(\boldsymbol{p}';\mathcal{P}',\mathcal{P})$ since the normal vector involved in its computation is from the corresponding one of $\mathcal{P}$. Note that terms similar to (3) are also used in (Wang et al., 2018; Tang et al., 2019) for single-view surface reconstruction. Our use of the term (3) in (4) is to encourage the consistency of local surface geometries between $\mathcal{P}'$ and $\mathcal{P}$, rather than to directly minimize (3) as in (Wang et al., 2018; Tang et al., 2019).

**The Combined Geometry-aware Objective** We use the following combined objective to learn the deformed $\mathcal{P}'$ by either adversarial or cooperative perturbations of individual points of $\mathcal{P}$

$$C_{Geometry}(\mathcal{P}',\mathcal{P}) = C_{\texttt{Chamfer}}(\mathcal{P}',\mathcal{P}) + \alpha \cdot C_{\texttt{Hausdorff}}(\mathcal{P}',\mathcal{P}) + \beta \cdot C_{\texttt{Curvature}}(\mathcal{P}',\mathcal{P}), \quad (5)$$

where $\alpha$ and $\beta$ are the weighting parameters.

## 3.2 GENERATION OF ADVERSARIAL POINT CLOUDS

Assume a point cloud classification model $f : \mathcal{X} \to \mathcal{Y}$. An adversarial example of a point cloud $\mathcal{P}$ is a crafted malicious input $\mathcal{P}'$ to the model $f(\cdot)$, with imperceptible deformation, such that $\mathcal{P}'$ is misclassified by the model. Let the true label of $\mathcal{P}$ be $y \in \mathcal{Y}$, it means that $f(\mathcal{P}') \neq y$. Adversarial examples can be generated either by untargeted or targeted attacks. Similar to (Xiang et al., 2019), we focus in this work on the more difficult task of targeted attack for point cloud data, which generates $\mathcal{P}'$ such that it is classified as a specified class $y' \neq y$, i.e., $f(\mathcal{P}') = y'$.

Among various attack models proposed for 2D image classification (Madry et al., 2017; Carlini & Wagner, 2017), we adopt the state-of-the-art framework of C&W attack (Carlini & Wagner, 2017). Its objective can be generally written as

$$\min_{\boldsymbol{x}'} C_{Adv}(\boldsymbol{x}') + \lambda \cdot \|\boldsymbol{x}' - \boldsymbol{x}\|_p,$$
(6)

where $\boldsymbol{x}$ is the benign signal (e.g., an image) and $\boldsymbol{x}'$ is the adversarial example to be optimized. The misclassification loss $C_{Adv}(\boldsymbol{x}')$ is regularized by a $\lambda$ weighted, $l_p$-norm based term that constrains the noise magnitude of adversarial $\boldsymbol{x}'$. To specify $C_{Adv}(\boldsymbol{x}')$, we assume a classification model $f(\cdot)$ be implemented as a deep network, and denote as $g(\cdot)$ the function that outputs the network logits, i.e., $g(\cdot)$ includes all layers of the network except the final softmax. Let the targeted label attacking $\boldsymbol{x}$ as $y'$, C&W attack commonly uses a margin based loss function as $C_{Adv}(\boldsymbol{x}') = \max\{\max_{i \neq y'} g_i(\boldsymbol{x}') - g_{y'}(\boldsymbol{x}'), 0\}$.

In this work, we adopt the C&W attack framework, and propose to replace the term of $l_p$-norm in (6) with our geometry-aware objective (5), in order to generate adversarial point clouds with imperceptible shape deformations, giving

$$\min_{\mathcal{P}'} C_{Adv}(\mathcal{P}') + \lambda \cdot C_{Geo}(\mathcal{P}',\mathcal{P}).$$
(7)

The original choice of margin based $C_{Adv}(\mathcal{P}')$ in C&W attack ceases pursuing more adversarial examples once $\max_{i \neq y'} g_i(\mathcal{P}') - g_{y'}(\mathcal{P}') \leq 0$, assuming that further optimization would reduce

the imperceptibility of the resulting $\mathcal{P}'$. Our proposed geometry-aware (5) allows us to take a more aggressive strategy, and we propose to use the following term as our misclassification loss

$$C_{Adv}(\mathcal{P}') = -\log\left(\exp(g_{y'}(\mathcal{P}'))/\sum_i \exp(g_i(\mathcal{P}'))\right). \tag{8}$$

Our proposed objective (7) with term (8) continues to pursue more adversarial, arguably less defendable, point clouds without introducing noticeable shape deformations, as empirically verified by our experiments in section 4.

### 3.3 GENERATION OF COOPERATIVE POINT CLOUDS

The adversarial objective (7) motivates us to think of an *opposite* direction of geometry-aware point cloud deformations that are less noticable by humans. Specifically, is it possible to perturb points of any $\mathcal{P}$ in a *cooperative, geometry-aware manner* such that the resulting $\mathcal{P}'$ is more favored by machine learning models in terms of improved classification confidence/accuracy? Technically, for a given classifier $f(\cdot)$, this seems to be achieved trivially by optimizing

$$\min_{\mathcal{P}'} C_{Coop}(\mathcal{P}') + \lambda \cdot C_{Geo}(\mathcal{P}', \mathcal{P}), \tag{9}$$

with the cooperative loss term as

$$C_{Coop}(\mathcal{P}') = -\log\left(\exp(g_y(\mathcal{P}'))/\sum_i \exp(g_i(\mathcal{P}'))\right), \tag{10}$$

where we simply use the true label $y$ of $\mathcal{P}$ to replace the targeted attack label $y' \neq y$ in (8).

Given knowledge of the true label $y$ of $\mathcal{P}$, the objective (9) seems only learn a deformed $\mathcal{P}'$ that overfits the classifier $f(\cdot)$, which could be practically less meaningful. In this work, we present the following interesting, but less explored investigation based on (9), which suggests that subtle deformations of shape instances of common object categories (e.g., those in ShapeNet (Chang et al., 2015)) could lead to practical meanings in terms of being better perceived by existing 3D point set classification models (Qi et al., 2017a;b; Wang et al., 2019b).

Assume we are given a set of $m$ point clouds $\{\mathcal{P}_i\}_{i=1}^m$ of different object categories. We take the following procedure on $\{\mathcal{P}_i\}_{i=1}^m$.

1. We divide $\{\mathcal{P}_i\}_{i=1}^m$ evenly into $s$ subsets, with consideration of class balance.
2. We use the first $s-1$ subsets as training data to train a classifier $f^s(\cdot)$, and use the objective (9) to cooperatively deform point clouds in the $s$th subset (the validation set) w.r.t. $f^s(\cdot)$.
3. We perform $s$ times of step 2 by using each of the $s$ subsets as the validation set, obtaining $f^i(\cdot)$, $i = 1, \ldots, s$, and the aggregated collection of deformed $\{\mathcal{P}'_i\}_{i=1}^m$.

The above procedure can optionally be conducted for multiple times to further increase the degree of deformations, with awareness of geometric imperceptibility. We use the obtained $\{\mathcal{P}'_i\}_{i=1}^m$ in a standard training-and-test setting of point set classification. That is, we split $\{\mathcal{P}'_i\}_{i=1}^m$ as training and test data, whose indices are the same as those of the original $\{\mathcal{P}_i\}_{i=1}^m$, and train a new classifier $\hat{f}(\cdot)$ to evaluate $\hat{f}(\cdot)$ on the split test data.

The obtained $\{\mathcal{P}'_i\}_{i=1}^m$ are indeed cooperative when performance of $\hat{f}(\cdot)$ is improved over that of the original classifier $f(\cdot)$. In fact, we have further conducted cross-model experiments by learning $\{\mathcal{P}'_i\}_{i=1}^m$ via the above procedure with PointNet (Qi et al., 2017a), and train and evaluate $\hat{f}(\cdot)$ of other representative point set classifiers (e.g., PointNet++ (Qi et al., 2017b) and DGCNN (Wang et al., 2019b)). We have also conducted experiments by converting the obtained $\{\mathcal{P}'_i\}_{i=1}^m$ to their mesh representations (Edelsbrunner et al., 1983), and uniformly re-sample points from the meshes to form $\{\widetilde{\mathcal{P}'}_i\}_{i=1}^m$; classification performance again improves by training and testing on $\{\widetilde{\mathcal{P}'}_i\}_{i=1}^m$, confirming that we are indeed cooperatively deforming the underlying object surfaces.

While our above investigations remain in the digital space, cooperative deformations of point clouds could be physically achieved either by changes of shape design for common rigid objects, or by designing special devices that actively control the reflections of laser pulses from LiDAR (Cao et al.,

2019). Physical-world adversarial attacks (Athalye et al., 2017; Eykholt et al., 2018) are actively pursued recently in the 2D image domain. Deformations of geometry-aware imperceptibility make sense here since we are not to change the geometric and/or topological surface structures of the objects; otherwise the deformations would have influence on human perception or functional attributes of the objects.

# 4 EXPERIMENTS

**Dataset** We use point clouds of object instances from ModelNet40 (Wu et al., 2015) to evaluate our proposed algorithms. The dataset consists of $12,311$ CAD models belonging to $40$ semantic categories. For each CAD model, $1,024$ points are uniformly sampled from its surface as the working point clouds, which are re-scaled into a unit ball following (Qi et al., 2017b).

**Models and Protocols** We evaluate the adversarial and cooperative generation of point clouds based on three representative classifiers, namely PointNet (Qi et al., 2017a), PointNet++ (Qi et al., 2017b), and DGCNN (Wang et al., 2019b). More details of the networks please refer to A.6. In our adversarial setting, we follow the official data split for 3D classification (Qi et al., 2017a;b) and use $9,843$ samples of point clouds for training classifiers that are to be attacked, and for testing, we follow (Xiang et al., 2019) and randomly select 25 samples from each testing set of the top-10 object classes (ordered by sample sizes of these classes, giving rise to *airplane, bed, bookshelf, bottle, chair, monitor, sofa, table, toilet, vase*); all adversarial attack algorithms are evaluated with a white-box, targeted attack protocol. In our cooperative setting, we set the subset number as $s = 5$.

**Evaluation Metrics** We respectively use the `attack success rate` (i.e., misclassification rate) and `classification accuracy` to evaluate the algorithmic effectiveness in our adversarial and cooperative settings. For both metrics, the higher, the better.

**Implementation Details** We set $k = 16$ to define local point neighborhoods for computation of normals and approximate curvatures in (3). We fix $\alpha = 0.1$ and $\beta = 1.0$ for our proposed geometry-aware loss (5), and the trade-off parameter $\lambda$ in (7) and (9) is optimized via 10-step binary search. We use Adam optimizer (Kingma & Ba, 2014) to train networks, and set its learning rate as $0.01$.

## 4.1 EVALUATION ON GENERATION OF ADVERSARIAL POINT CLOUDS

In this section, we evaluate the efficacy of our proposed geometry-aware objective (7) for generation of adversarial point clouds. We dub our method as GeoAwareAdv and compare with the method (Xiang et al., 2019), which is among the only few existing works (Liu et al., 2019a;b) addressing adversarial point clouds and takes the same white-box, targeted attack as our method does. Objective of (Xiang et al., 2019) strictly follows the C&W attack (6) (Carlini & Wagner, 2017), with a margin-based misclassification loss regularized by $l_2$-norm constraining magnitude of point perturbations. We also compare with a degenerate version of our method, dubbed GeoDegenerateAdv, which replaces the geometry-aware term (5) with a similar $l_2$-norm as in (Xiang et al., 2019). These experiments are conducted using PointNet (Qi et al., 2017a) as the model of classifier.

**Ablation Studies** To investigate how different terms in our proposed geometry-aware loss (5), we conduct ablation studies by removing each of them from (5), and use the remaining ones into the adversarial objective (7) for point cloud generation. Figure 1) shows that these terms all together contribute to smooth and outlier-free results. Without using CD, the shape deformations go to unexpected twists. The use of HD and curvature terms largely removes generation of outliers, with the later one further improving the surface smoothness.

**Comparative Results** We report comparative results of our GeoAwareAdv, GeoDegenerateAdv, and the method (Xiang et al., 2019) in Table 1 and Figure 2. Table 1 compares the attack success rates under two defense methods of SOR (Zhou et al., 2018) and Random Romoval (RR). SOR is the current state-of-the-art method to defend attacking of point clouds, which works by dropping certain points from a point cloud based on statistical analysis. RR simply drops points from a point cloud randomly. Based on our stronger adversarial term (8), both of our methods achieve better success rates than those of (Xiang et al., 2019) under different dropping settings of the defenses SOR and RR. More significantly, the stronger attacking effects of our GeoAwareAdv are achieved without introduction of noticeable outliers and surface irregularities, as shown in Figure 2; in contrast, the

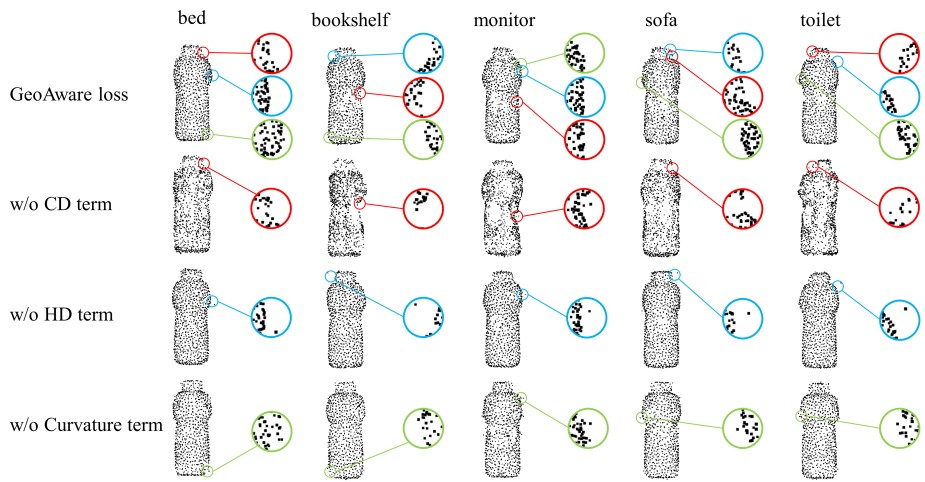

Figure 1: Ablation studies on the effectiveness of different terms in our geometry-aware loss (5) for generation of adversarial point clouds. See the main text for experiment settings of these results. More quantitative results please refer to A.4.

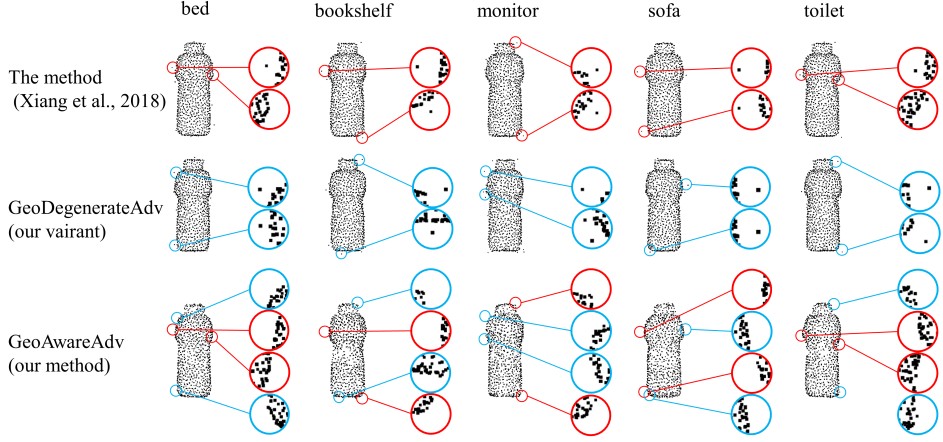

Figure 2: Comparative quality results of different methods successfully attacking the classifier of PointNet (Qi et al., 2017a).

use of simple $l_2$ norm in (Xiang et al., 2019) produces its adversarial results with clear outliers. Advantages of our GeoAwareAdv are essentially due to the use of HD and curvature terms in our geometry-aware loss (5).

Table 1: Success rates (%) of different methods attacking the classifier of PointNet (Qi et al., 2017a) under defenses of SOR (Zhou et al., 2018) and Random Romoval (RR). We report results with different numbers of dropping points for the two defense methods. For all methods, attack success rates are $100\%$ without defense. More details of RR and SOR please refer to A.7.

| Method | Defense method | Attack success rate (%) defense by dropping different numbers of points | | | | | | | | |
|---|---|---|---|---|---|---|---|---|---|---|
| | | 1 | 2 | 4 | 8 | 16 | 32 | 64 | 128 | 256 |
| The method (Xiang et al., 2019) | | 91.78 | 87.02 | 80.71 | 70.44 | 58.31 | 47.07 | 34.89 | 23.82 | 16.00 |
| GeoDegenerateAdv (our vairant) | RR | 98.17 | 96.63 | 94.61 | 90.06 | 84.89 | 76.65 | 61.32 | 38.64 | 17.38 |
| GeoAwareAdv (our method) | | **99.87** | **99.78** | **99.60** | **99.38** | **98.57** | **97.77** | **94.65** | **79.97** | **47.48** |
| The method (Xiang et al., 2019) | | 12.71 | 7.29 | 3.69 | 1.96 | 1.51 | 1.24 | 0.71 | 0.67 | 1.56 |
| GeoDegenerateAdv (our vairant) | SOR | 29.37 | 13.73 | 6.02 | 3.25 | 1.83 | 0.94 | 0.89 | 0.80 | 1.25 |
| GeoAwareAdv (our method) | | **98.84** | **97.86** | **96.08** | **92.74** | **87.62** | **77.06** | **61.29** | **41.38** | **28.20** |

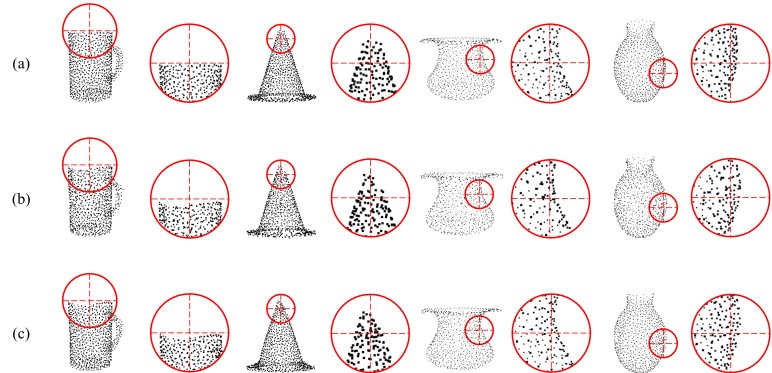

Figure 3: Visualization of cooperative point clouds generated based on PointNet. (a) Vanilla point clouds, (b) Cooperative point clouds, and (c) Uniformly re-sampling on the mesh surface of (b).

## 4.2    EVALUATION ON GENERATING GEOMETRY-AWARE COOPERATIVE POINT CLOUDS

Table 2: Classification accuracy (%) of different networks trained on different source of data, including vanilla point clouds, cooperative point clouds and its variant generated based on PointNet. The state-of-the-art Geo-CNN (Lan et al., 2019) achieves the classification accuracy of 93.90%.

| Data | Classification accuracy (%) | | |
| --- | --- | --- | --- |
|  | PointNet | PointNet++ | DGCNN |
| Vanilla point clouds | 87.96 | 89.50 | 90.64 |
| Cooperative point clouds (ours) | 96.88 | 94.32 | 96.27 |
| Re-sampling variant (ours) | 94.81 | 92.78 | 94.12 |

**Comparative Evaluation** Cooperative point clouds are generated based on the PointNet via optimizing objective function (9), and a variant can be re-sampled point clouds from its mesh surface. A number of recent classifiers are trained and evaluated on vanilla and cooperative point clouds respectively, whose results are reported in Figure 3 and Table 2. It can be observed from Table 2, both cooperative point clouds significantly outperform the vanilla ones by a large margin, *i.e.,* increase at least 6% on classification accuracy. Moreover, performance gap between two types of cooperative point clouds can be explained by both approximation errors of mesh reconstruction and the changes on point distribution, which encourage us to consider adversarial shape deformations on the mesh level.

**Cross-model experiments** We conduct one more cross-model experiment to test transferable characteristics of cooperative point clouds across classifiers, *i.e.,* generating cooperative examples on PointNet, while train and evaluate other classifiers. Results in Table 2 show consistent improvement on classification accuracy achieved for different neural classifiers, which reveal that cooperative shape deformations on point clouds can capture discriminative geometric patterns favor for semantic object classification, which are independent on neural models.

## 5    CONCLUSIONS

In this paper, we propose a compact geometry-aware loss to constrain point-wise imperceptible perturbations to achieve similar geometric smoothness and fairness of local patches and their global topological configuration in deformed point clouds as the original ones. Experiment results reveal the rationale of three components in the proposed geometry-aware loss for generating adversaries without evident outliers and shape irregularities, which can thus achieve adversarial effects even confronting the SOR defense. Moreover, cooperative generation on point clouds also demonstrates its positive effects on improving classification with aware of geometric properties.

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

# A APPENDIX

## A.1 VISUALIZATION OF MORE GEOMETRY-AWARE ADVERSARIAL EXAMPLES.

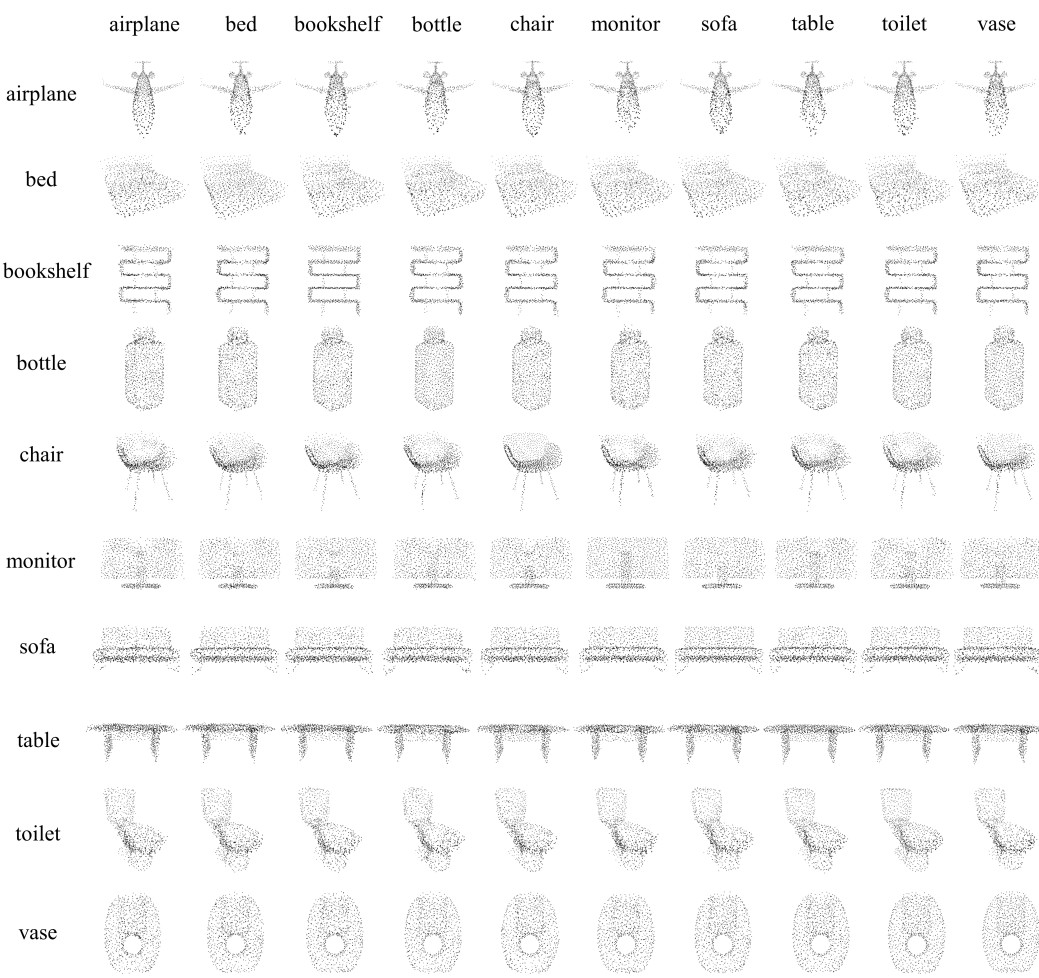

Figure 4: Visualization of our proposed geometry-aware adversarial examples based on PointNet. We show all kinds of our selected categories here targeted on all the other classes. The diagonal examples are corresponding benign ones.

## A.2 VISUALIZATION OF MORE GEOMETRY-AWARE COOPERATIVE EXAMPLES.

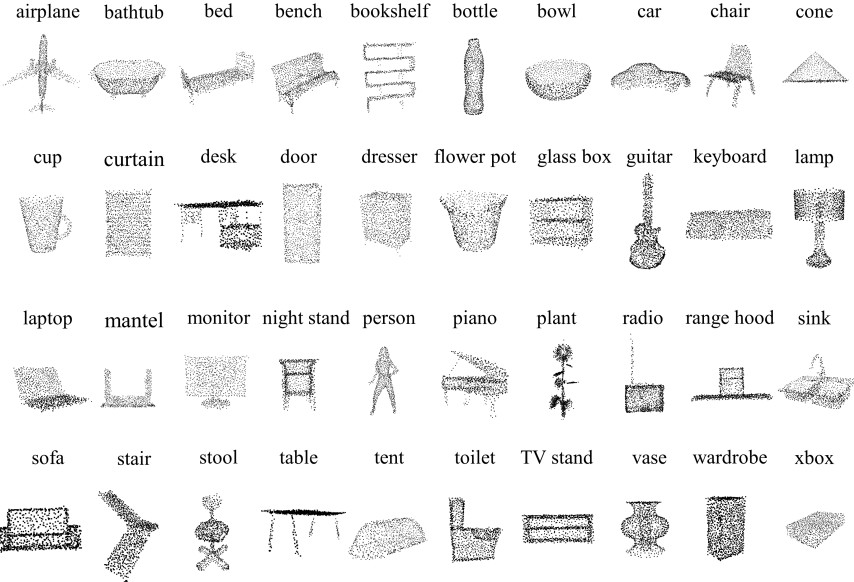

Figure 5: Visualization of our proposed geometry-aware cooperative examples based on PointNet. We show all kinds of categories in the dataset.

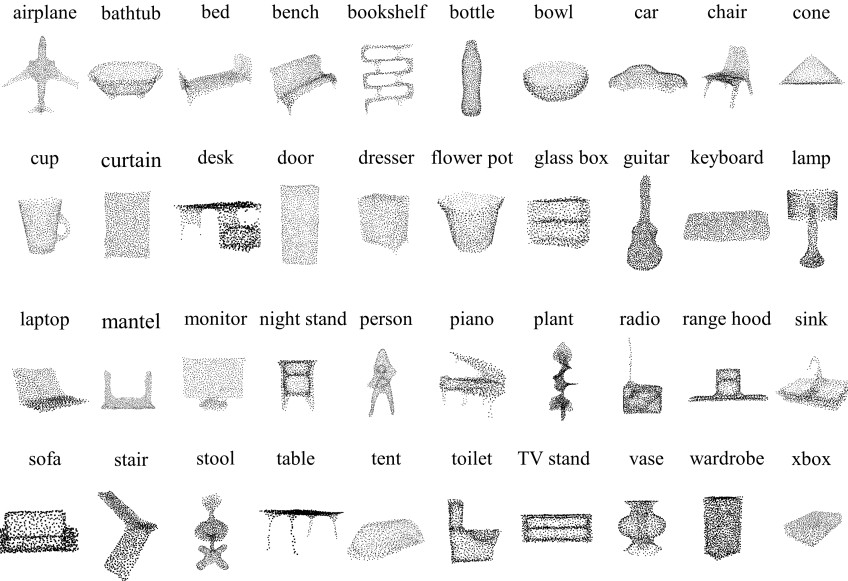

Figure 6: Visualization of our proposed resampling-after-reconstruction on geometry-aware cooperative examples based on PointNet. We show all kinds of categories in the dataset.

## A.3 ABLATION STUDY OF DIFFERENT $\alpha$ AND $\beta$.

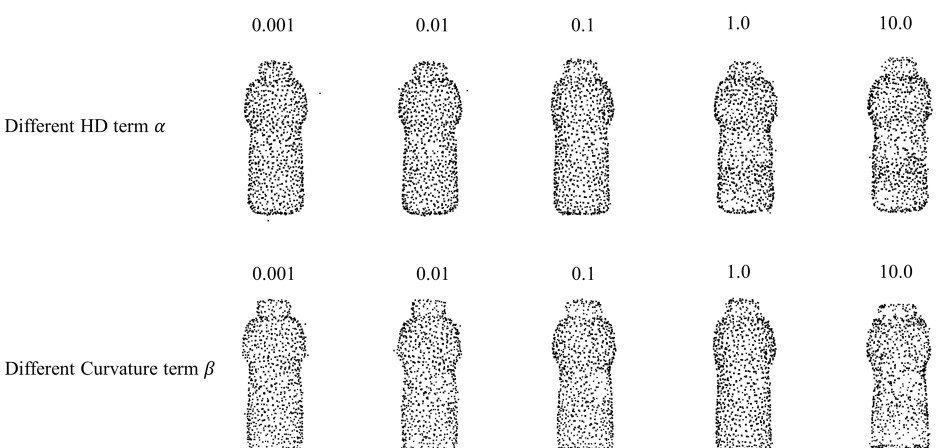

Figure 7: Qualitative results of adopting different $\alpha$ and $\beta$. All the adversarial examples are generated via targeted attacked method to the same target. We fix all the other hyperparameters when tuning one. The default parameters are $\alpha = 0.1$ and $\beta = 1.0$. As can be seen in the figure, smaller $\alpha$ leads to more obvious outliers; and larger $\alpha$ leads to the deformation of the point cloud, due to the overwhelming focus on the outliers while ignoring the whole shape. On the other hand, smaller $\beta$ leads to high frequency outliers near the surface; and again, larger $\beta$ leads to the deformation of the whole point cloud.

## A.4 QUANTITATIVE RESULTS FOR FIGURE 1.

It's hard to define a metric evaluating geometrical smoothness of a point cloud directly due to its irregularity and lacking of geometric topological connection. Hence, we introduce an approximation metric mainly focuses on measuring the strength of isolated noisy points and the roughness of local patches, which can be defined as:

$$S(\mathcal{P}) = \max_{\boldsymbol{p} \in \mathcal{P}} \sum_{\boldsymbol{q} \in \mathcal{N}_{\boldsymbol{p}}} D(\boldsymbol{q}, \boldsymbol{T}\mathcal{N}_{\boldsymbol{p}}) \tag{11}$$

where estimated tangent plane of the $k$ nearest neighbors $\mathcal{N}_{\boldsymbol{p}}$ of $\boldsymbol{p}$ is denoted by $\boldsymbol{T}\mathcal{N}_{\boldsymbol{p}}$, and $D$ denotes the distance function between the points and the tangent plane, where we use $l_2$ norm in our settings. Intuitively, we are calculating the residual of first-order Taylor expansion of the points  we estimate the first-order approximation of a point $\boldsymbol{p}$, i.e., the tangent plane $\boldsymbol{T}\mathcal{N}_{\boldsymbol{p}}$, and then measures how far its neighbor points $\mathcal{N}_{\boldsymbol{p}}$ are to this plane. Note that we set the $|\mathcal{N}_{\boldsymbol{p}}| = k$ to be small to reduce approximation error, where in the following settings $k = 16$.

We evaluate $S(\mathcal{P})$ on the experiments of Figure 1, and the quantitative results are shown in the following Table 3. As can be seen in the table, with all the losses, the geometry-aware adversarial point clouds achieve the lowest $S(\mathcal{P})$, where the point clouds are smoothest. And with many outliers, the adversarial point clouds without Hausdorff Distance get the highest $S(\mathcal{P})$. The quantitative results of the point clouds without Chamfer Distance are a little higher than the point clouds without Consistency of Local Curvatures, which is consistent with our perception  the point clouds without Consistency of Local Curvatures are also very smooth expect for some high frequency outliers near their surfaces.

Table 3: Quantitative results of the ablation study of Figure 1. We evaluate smoothness metric S(P) with and without individual components. We take average over all the samples. Lower value means smoother point clouds.

| Methods | Smoothness avg. |
|---|---|
| Whole geometry-aware loss | **0.1047** |
| Without Chamfer Distance | 0.1214 |
| Without Hausdorff Distance | 0.1895 |
| Without Consistency of Local Curvatures | 0.1210 |

## A.5 USER STUDY ON AMAZON MECHANICAL TURK (AMT).

We conducted a user study on Amazon Mechanical Turk (AMT) in order to verify the imperceptible quality of our adversarial examples. We upload the snapshots of three kinds of point clouds including the benign point clouds, the adversarial point clouds generated via the method of Xiang et al. (2019) and our geometry-aware adversarial ones. All the adversarial point clouds can successfully mislead PointNet. Participants were asked to compare which one of the adversarial point clouds are more geometrically similar to the original one. The order of two kinds of adversarial point clouds was randomized and all the images appeared in the middle of the screen on each trial. Each participant could conduct at most 30 trials and each adversarial images can be shown to 50 different participant at most. In total, we conduct 1500 trials among 128 participants. And our geometry-aware adversarial point cloud are considered to be closer to the original ones in 82.06% of the trials, which indicates that our geometry-aware point clouds are more imperceptible. We think this experiment can further support our opinion that our geometry-aware adversarial examples are more imperceptible to humans. And we will attach this experiments to the appendix of our revised paper.

## A.6 INTRODUCTION OF OUR POINT CLOUD CLASSIFIER.

The PointNet (Qi et al., 2017a) and Pointnet++ (Qi et al., 2017b) are the first attempts to explore deep point cloud classification, with the permutation invariance of points in multi-layer perceptrons (MLPs) and a symmetric function for aggregating features. Both methods can only implicitly model global semantic patterns from 3D geometry. Another groups of algorithms design graph based convolution operation on irregular distributed structure of points such as DGCNN (Wang et al., 2019b). In DGCNN, an edge convolution operation is proposed on a dynamic graph to discover local geometric manifolds.

We mainly focus on PointNet Qi et al. (2017a) as our baseline due to its simple structure and good performance. The classier $f$ can be formulated as $f(\boldsymbol{P}) = \gamma(\max_{\boldsymbol{p}_i \in \boldsymbol{P}}\{h(\boldsymbol{p}_i)\})$, where $\gamma$ and $h$ are two learnable parameters of the neural network.

## A.7 INTRODUCTION OF DEFENSIVE ALGORITHM

We adopt two kinds of defense method including random removal method (RR for short) and statistic outlier removal method (SOR for short)Zhou et al. (2018) as the defensive method.

In RR, we randomly select a subset of points from the point cloud and drop it. And in SOR, we calculate the average distance $d_{\boldsymbol{p}}$ of a point $\boldsymbol{p}$ to its $k$-nearest neighbors, which can be denoted by

$$d_{\boldsymbol{p}} = \frac{1}{k} \sum_{\boldsymbol{q} \in \mathcal{N}_{\boldsymbol{p}}} \|\boldsymbol{p} - \boldsymbol{q}\|_2 \tag{12}$$

And then we calculate the mean $\bar{d}$ and standard deviation $\sigma_d$ over all the distance $d_{\boldsymbol{p}}$ of points in the point cloud $\boldsymbol{p} \in \mathcal{P}$. We remove all the points that fall outside $\bar{d} \pm a \times \sigma_d$, where $a$ is set to be 1.1 in Zhou et al. (2018). According to our experiments, around 100 points will be dropped out of 1024 points when $a$ is set to be 1.1.

However, in our settings, we drop a fix number points with $m$-largest $d_{\boldsymbol{p}}$ for fairer comparison between different methods. We drop $k = 1, 2, 4, 8, 16, 32, 64, 128, 256$ respectively in Table 1.

