# OpenReview forum: "Geometry-aware Generation of Adversarial and Cooperative Point Clouds"
_ICLR.cc/2020/Conference — Reject_

### Official Review · AnonReviewer3 · 2019-10-22
**Official Blind Review #3**

**Rating:** 3

**Review:**

This paper proposes a novel loss function to account for imperceptible, geometry-aware deformations of point clouds. The loss is used in two cases: generating adversarial point clouds to attack representative models of point set classifiers, and generating cooperative point clouds to improve classification confidence or accuracy. The combined geometry-aware objective is well-introduced, which mainly contains Chamfer distance term, Hausdorff distance term and local curvatures consistency term. The authors apply the geometry-aware objective to generate adversarial point clouds by adopting the framework of C&W attack. For generating cooperative point clouds, the authors introduced a training procedure to reduce the overfitting of the deformed point clouds. Most of the experiments are well-conducted, and demonstrate the effectiveness of the proposed loss function.

Overall, the paper is a well-written and could be an interesting contribution. However, I would like it better if it does not contain the part about the cooperative point clouds, which are not very motivated and the experiment settings are not very convincing. The algorithm in page 5 inherently used testing labels (suppose some testing data is in the training split of one of those cross-validation folds, training for the P_i' for the rest of the training data). I don't know why experiment results generated by this method would be of any meaning. I would be OK to accept this paper if the cooperative point cloud part is dropped.

- In page 5, the authors introduce a procedure to train the model. The first step would require take the “consideration of class balance”. It would be better to give some details on how the class balance is considered.

- In ablation study, the authors give some visualization results in Figure 1. However, it would be more interesting if the authors could give some quantitative results.

- It would be interesting to show some ablation study on how to choose the  α, β and λ .


**Experience Assessment:**

I have published one or two papers in this area.

**Review Assessment: Checking Correctness Of Derivations And Theory:**

I assessed the sensibility of the derivations and theory.

**Review Assessment: Checking Correctness Of Experiments:**

I carefully checked the experiments.

**Review Assessment: Thoroughness In Paper Reading:**

I read the paper thoroughly.

---

> ### Author Response · Authors · 2019-11-14
> **Re: Official Blind Review #3**
>
> Thank you for your constructive comments. We have improved the paper based on these comments. Our responses to individual comments are as follows.
>
> Q1. What’s the significance of “cooperative part” in the paper?
>
> Reply: We appreciate the reviewer's comment. We agree that the cooperative part of this paper is of exploratory nature; however, we want to take this rebuttal opportunity to better explain this exploratory attempt. Our cooperative point clouds can be practically meaningful in terms of guiding design of certain objects such that they can be better perceived by in-built machine learning systems in physical world. As shown in Figure 3 in the paper, the deformations are subtle, which has little influence on human perception or functional attributes of the objects while being physically achievable. And such subtle modifications on the objects lead to a significant gain in the classification accuracy. Therefore, we can fabricate objects in these shapes! For example, if we learn a subtle shape modification that would improve the accuracy of car classification, such guidance can be used for designing auto-driving cars. This could be a way to achieve safe auto-driving, complementary to that of designing better machine learning models. We note that the transferability of cooperative examples mentioned in our paper makes it possible to do so, because all the classifiers gain accuracy boosts no matter from which the cooperative shapes are generated. The quantitative results can be found in Table 2 of our paper and the details of this experiment can be found in “Cross-model experiments” part of Section 4.2.
>
> We believe that the cooperative part is an interesting exploratory attempt, possibly making real-world objects more suitable for machine learning systems. We expect our initial attempt cloud encourage more discussions in the community about object geometries favorable for classification.
>
> However, we are also willing to delete this part if all the reviewers and area chairs reach a consensus.
>
> Q2. What’s the meaning of “consideration of class balance” in Page 5?
>
> Reply: Many thanks for the suggestion. “Class balance” here indicates that we randomly and uniformly split the instances of each class into a fixed fold size (e.g., 5 in our experiments), and the instances of all classes in each fold keep the same data distribution as that of the whole dataset.
>
> Q3. More quantitative results of ablation study is needed.
>
> Reply: Many thanks for this interesting suggestion. It's hard to define a metric evaluating geometrical smoothness of a point cloud directly, due to its irregularity and lack of topological connection. Hence, we introduce an approximation metric S(P) mainly focusing on measuring the strength of isolated noisy points and the roughness of local patches. More details of the metric please refer to  Appendix A.4 in the revised paper.
>
> We evaluate S(P) on the experiments of Figure 1 in our paper, and the quantitative results are shown in the following Table 3 in the revised paper. As can be seen in the table, with all the losses, the geometry-aware adversarial point clouds achieve the lowest S(P), where the point clouds are smoothest. And with many outliers, the adversarial point clouds without Hausdorff Distance get the highest S(P). The quantitative results of the point clouds without Chamfer Distance are a little higher than the point clouds without Consistency of Local Curvatures, which is consistent with our perception – the point clouds without Consistency of Local Curvatures are also very smooth except for some high frequency outliers near their surfaces
>
> Q4. It would be interesting to show some ablation study on how to choose the alpha, beta and lambda.
>
> Reply: Many thanks for the suggestion. We performed an ablation study on different alpha and beta and attach the quality results to the Appendix A.3 of our revised paper. As shown in the figure, smaller alpha values (less attention to Hausdorff Distance) lead to more obvious outliers; and larger alpha values (more attention to Hausdorff Distance) lead to the deformation of the point cloud, due to the overwhelming focus on the outliers while ignoring the whole shape. On the other hand, smaller beta values (less attention to consistency of local curvatures) lead to high frequency outliers near the surface; and again, larger beta values (more attention to consistency of local curvatures) lead to the deformation of the whole point cloud. We do not perform the ablation study on the hyper-parameter of lambda (the balance parameters of adversarial term and imperceptibility term) because we have adopted the strategy of binary-search on it in all of our experiments. During the binary-search, lambda is self-adjusted adaptively: if the adversarial attack is successful, lambda will become larger, and vice verse. And hence, it’s a dynamically adjusted parameter already.

---

### Official Review · AnonReviewer2 · 2019-10-24
**Official Blind Review #2**

**Rating:** 3

**Review:**

This paper looks at the task of (adversarially or cooperatively) perturbing a point cloud in context of a classification task. It follows the prevailing paradigm of changing the original input in the direction of a high positive/negative gradient while staying ‘close’ to the original input, and the key contribution here is to define a different notion of ‘closeness’.

While previous work (Xiang et. al., at least for ‘point addition attack’) used a combination of chamfer and Hausdorff distance as the notion of closeness, this paper additionally includes change in curvature (which is an intuitive term to include) when computing the adversarial/cooperative updates to the point cloud. The obtained results do visually look less perturbed compared to the previous approach, and the obtained adversarial shapes are more robust against two defenses studied.

Concerns/Questions:

1) If a point cloud P’ is only a (small) perturbation of a point cloud P, then the Chamfer distance / Hausdorff distance is essentially the  L_2 / L_infinity norm of their difference. While the use of curvature terms is different, I feel the claims of importance of using ‘distance metric of point clouds’ is not very well justified (as I’d expect essentially same results if these two terms were replaced by L2 and L_infinity norms instead). I think the use of these terms was more necessary in the work of Xiang et. al., as they allowed point addition, so the ‘norm of difference of point clouds’ is not well defined.

2) This paper uses a different (more aggressive) adversarial term (in Eqn. 8) compared to Xiang et. al., so it is not surprising that the results in Table 1 indicate more robustness to defenses.

3) In addition to the above comments about specifics, I feel this work’s contribution over prior work is not significant. While Xiang et. al. did use L2 norm for their perturbation case, they did investigate the Chamfer/Hausdorff distances for another scenario, and therefore the main contribution here is an additional loss term.

4) This is perhaps a hard concern to address, but simply showing some qualitative results to highlight that the changes are ‘imperceptible’ is not sufficient. Ideally, one should report a curve on ‘change perceptibility’ vs ‘attack success rate’ (though this would require some notion of perceptibility that was not used in optimization). Alternately, one could compare methods via A/B testing on mechanical turk, asking ‘Which are these two shapes are closer to the original one?’, and ablate for a certain level of confidence on the wrong class, which approach led to less changes. The current results simply show some examples, but provide no empirical way of judging which approach actually leads to more imperceptible changes.

Overall, though the results are perceptually encouraging, I have slight concerns the empirical results reported. However, the primary issue is that the contribution regarding the additional term, while intuitive, is not a significant one in its own right.

While the rating here only allows me to give a ‘3’ as a weak reject, I am perhaps a bit more towards borderline (though leaning towards reject) than that indicates.


**Experience Assessment:**

I have read many papers in this area.

**Review Assessment: Checking Correctness Of Derivations And Theory:**

I assessed the sensibility of the derivations and theory.

**Review Assessment: Checking Correctness Of Experiments:**

I carefully checked the experiments.

**Review Assessment: Thoroughness In Paper Reading:**

I read the paper thoroughly.

---

> ### Author Response · Authors · 2019-11-14
> **Re:  Official Blind Review #2**
>
> Thank you for your constructive comments. We have improved the paper based on these comments. Our responses to individual comments are as follows.
>
> Q1. Is Chamfer distance / Hausdorff distance essentially the L_2 / L_infinity norm?
>
> Reply: We emphasize that the nature of Chamfer / Hausdorff distance is fundamentally different from that based on L_2 / L_infty norm; the former measures the distance between point clouds when taking a point cloud as a whole, and the latter measures the distance of corresponding points between point clouds. Consequently, small point cloud perturbations under the two distance metrics give fundamentally different adversarial results. With Chamfer / Hausdorff distance, individual points of an input point cloud could drift far away along the object surface from their original positions while maintaining the same underlying object surface as the original one does. This would never happen when perturbation is measured by L_2 / L_infty norm of individual points.
>
> It is essentially our use of Chamfer / Hausdorff distance based metrics that make it possible to produce adversarial point clouds that have larger freedoms of local surface deformations at smaller magnitudes of perturbation measured globally. However, such increased local freedoms bring new issues related to geometric regularities of local surfaces, thus motivating us to consider the additional curvature based loss to compensate for.
>
> Q2. How dose the more aggressive adversarial term affect the results?
>
> Reply: We have done the experiments of replacing the adversarial term of C1 method with the identical adversarial term in Eqn. (8) of our paper, which is named as “GeoDegenerateAdv”. It can be regarded as a transitive method from C1 method to our geometric-aware method, i.e., the only difference between “GeoDegenerateAdv” and “The method (Xiang et. al., 2019)” is the former adopts a more aggressive adversarial term. All the results are shown in Table 1 in the paper.
>
> As the reviewer might expect, the “GeoDegenerateAdv” achieves slightly better robustness against RR / SOR defense method compared to the method of C1. And as shown in Figure 2, outliers in “GeoDegenerateAdv” are more noticeable, e.g. in the “monitor” case. Thus, it is not surprising that it cannot defense SOR method. And in most cases, our "GeoAwareAdv" outperforms the others significantly. This result indicates that though there is improvement brought by such aggressive, adversarial term, it is not the key of making our geometry-aware adversarial point clouds more robust. Our geometry-aware adversarial method achieves stronger attack by avoiding noticeable outliers and surface irregularities, i.e., we spread the adversarial information into the whole point set instead of some particular outliers. Therefore, the statistic-based defensive method of RR / SOR loss their effects.
>
> Q3. What's the contribution of this work over prior works?
>
> Reply: What we want to highlight in this paper is that no matter how we construct an adversarial example, it should maintain the implied attribute itself, e.g., the geometry properties in 3D point clouds. Adversarial examples have attracted much attention due to the fascinating phenomenon that the perturbations on them are so imperceptible that makes no sense should any classifier mis-classify them.
>
> We note that the notions of imperceptibility would change when applying adversarial attacks to different data domains.  It is of essential importance to first define a clear notion of imperceptibility since as long as the perturbations are large enough, the obtained samples will cross the decision boundaries.
>
> In the present paper, we are trying to answer the question that “can we generate imperceptible adversarial point clouds?”. This problem is non-trivial because humans are very sensitive to shape changes. Existing works on adversarial point clouds mainly focus on adversarial effects of mis-classification, and our work makes the first attempt to perturb point clouds with the awareness on local geometric properties and inhabitation of evident outliers. Based on that, we proposed several losses to construct geometry-aware adversarial point clouds. Not only have we succeeded in generating more imperceptible adversarial point clouds, but also our adversarial point clouds perform much better when encountering countermeasures as a bonus.
>
> Q4. More qualitative results highlighting that the deformations are ‘imperceptible’ is needed.
>
> Reply: Thanks for this interesting suggestion. During this rebuttal period, we conducted a user study on Amazon Mechanical Turk (AMT) in order to verify the imperceptible quality of our adversarial examples. We have included these results of user study in Appendix A.5 in the revised paper.
>
>
> C1: Xiang, Chong, Charles R. Qi, and Bo Li. "Generating 3d adversarial point clouds." Proceedings of the IEEE Conference on Computer Vision and Pattern Recognition. 2019.

---

### Official Review · AnonReviewer1 · 2019-10-26
**Official Blind Review #1**

**Rating:** 8

**Review:**

This paper describes a new targeted adversarial attack against 3D point cloud object classifiers that is robust to several countermeasures. The attack finds a point of the target class that is close to the original point cloud in terms of a more complicated metric that combines the Hausdorff distance, the Chamfer distance, and a curvature distance measure. The proposed attack is 100% successful against several different state of the art classifiers on a dataset of 1024-point clouds sampled from 25 instances of CAD models of each of 10 common objects without any countermeasures.  When the Random Removal countermeasure is used, the attack is still successful almost 50% of the time even when 256 points are removed as compared to two other attacks that are only ~17% successful. When the SOR countermeasure is used, the attack is 60% successful when 64 points are removed as compared to <1% for the comparison attacks. The attack can also be used in reverse for data augmentation in training and can cut error rates almost in half.

Overall, this seems like a large improvement over current approaches. The paper does a good job of explaining its approach and motivation and does an excellent job of situating its contributions within the existing literature. The experiments show that the improvements in success are large over competing attacks, that they are robust to current countermeasures. They also show that when used "cooperatively" they can improve performance substantially.

One issue that could be easily addressed is that Table 2 is mentioned on page 5, but doesn't appear until the end of page 7. It is also mentioned before Table 1. So I would recommend changing it to Table 1 and introducing it before it is mentioned.

It would also be nice to have some basic descriptions of the point cloud classifiers and the SOR countermeasure. Space for this could be regained from some of the figures, which are informative, but over-represented.

**Experience Assessment:**

I have read many papers in this area.

**Review Assessment: Checking Correctness Of Derivations And Theory:**

I assessed the sensibility of the derivations and theory.

**Review Assessment: Checking Correctness Of Experiments:**

I assessed the sensibility of the experiments.

**Review Assessment: Thoroughness In Paper Reading:**

I read the paper at least twice and used my best judgement in assessing the paper.

---

> ### Author Response · Authors · 2019-11-14
> **Re: Official Blind Review #1**
>
> Thank you for your constructive comments. We have improved the paper based on these comments. Our responses to individual comments are as follows.
>
> Q1. One issue that could be easily addressed is that Table 2 is mentioned on page 5, but doesn't appear until the end of page 7. It is also mentioned before Table 1. So I would recommend changing it to Table 1 and introducing it before it is mentioned.
>
> Reply: We appreciate the suggestion. We have tried to follow the suggestion and change the positions of Table 1 and Table 2 accordingly. But we found that it didn’t conform to our original narrative order that first introduces geometry-aware adversarial point clouds and then introduces cooperative ones. We agree that it would raise inconvenience when mentioning Table 2 on page 5 without showing it until the end of page 7. We thus revised the paper by referring Table 2 at appropriate positions.
>
> Q2. It would also be nice to have some basic descriptions of the point cloud classifiers and the SOR countermeasure. Space for this could be regained from some of the figures, which are informative, but over-represented.
>
> Reply: Thanks for the suggestion. In the revised paper, we introduce the point cloud classifiers in Appendix A.6 and the defense method in Appendix A.7.

---

### Decision · Program_Chairs · 2019-12-19

**Decision:**

Reject

**Comment:**

This paper offers an improved attack on 3-D point clouds. Unfortunately the clarity of the contribution is unclear and on balance insufficient for acceptance.